# Organelle Genomes of *Nardostachys jatamansi* Offer New Perspectives into the Evolutionary Dynamics of Caprifoliaceae

**DOI:** 10.3390/biology14091219

**Published:** 2025-09-08

**Authors:** Yanli Xiong, Yi Xiong, Qingqing Yu, Xiao Ma, Xiong Lei

**Affiliations:** 1Sichuan Academy of Grassland Sciences, Chengdu 611700, China; yanlimaster@126.com (Y.X.); yuqinggzu93@126.com (Q.Y.); 2College of Grassland Science and Technology, Sichuan Agricultural University, Chengdu 611130, China; xiongyi95@126.com

**Keywords:** organelle genome, *N. jatamansi*, IGT, ecological niche

## Abstract

The restricted availability of plant organelle genomes poses a challenge to our complete understanding of the structural complexities of the mitochondrial genome and the dynamics of gene transfer between chloroplasts and mitochondria. While the Caprifoliaceae family is significant for its diverse cash crops and ornamental plants, only the mitochondrial genome of one species within this family has been sequenced. This study reveals the organelle genomes of a notable Tibetan medicinal plant, *Nardostachys jatamansi*, and encompasses a comparative analysis of the chloroplast genome within the Caprifoliaceae family.

## 1. Introduction

Photosynthesis and respiration are fundamental mechanisms that underpin plant biomass accumulation. These vital physiological processes are regulated by chloroplasts and mitochondria. Investigations into the genomes of these two organelles (chloroplast and mitochondrial genomes) have provided significant insights into their evolutionary history and functional roles [1]. Nevertheless, a comprehensive understanding of their structural dynamics and non-coding regions remains incomplete. This is primarily due to the scarcity of plant mitochondrial sequencing efforts, compound by the limited sequencing of organelle genomes from only 285 species to date [2]. Out of the restricted pool of sequenced mitochondrial species, a mere 34.4% pertain to angiosperms. This disparity could stem from challenges in precise assembly, particularly when compared to the relatively more straightforward assembly processes associated with moss and green algal mitochondrial genomes. Interestingly, the variations in structural complexity of mitochondrial genome could be associated with seed evolution [2]. Seed hydration and dehydration are essential metabolic processes, these processes require efficient double-strand break repair, which is closely linked to repetitive sequences. On the other hand, the phenomenon of intracellular gene transfer (IGT) is an intriguing occurrence noted in plant organelle genomes. Continuous IGT has the potential to continually modify the plant genome, consequently enhancing its complexity [3].

*Nardostachys jatamansi* (D.Don) DC. is a perennial herb belonging to the genus *Nardostachys* within the Caprifoliaceae family. It is a commonly employed Tibetan medicinal plant whose dried roots are utilized in pharmaceuticals [4]. These roots are believed to have therapeutic properties, including the regulation of metabolism, mitigation of pain, alleviation of stagnation and enhancement of spleen function, reduction in dampness and swelling when applied externally, and the potential to treat ailments such as loss of appetite, abdominal bloating, and vomiting [5]. Beyond its therapeutic applications, it serves as an ingredient in Tibetan incense and has the potential to be incorporated into various daily chemical products and food items [6]. The valuable medicinal properties of *N. jatamansi* have led to excessive mining if its wild resources, resulting in habitat degradation and interference from other organisms [7]. As a consequence, *N. jatamansi* has been listed as a second-class protected Tibetan medicine plant in China [8]. Furthermore, its conservation has been recognized internationally, with its inclusion in the IUCN Red List of Endangered Species [9,10]. Nevertheless, the scarcity of reference genome information for *N. jatamansi* has limited molecular-level studies. Existing research predominantly centers on the summarization and analysis of its pharmacological properties, chemical composition, clinical application, and related aspects [11,12]. This significantly hinders the comprehensive understanding of the genetic diversity of *N. jatamansi* and the development of corresponding conservation strategies. Previous research has employed chloroplast and mitochondrial markers (e.g., SSRs, genes, or intergenic regions) to explore genetic diversity in diverse angiosperms and related species like *Camellia huana* [13], *Elymus sibiricus* [14], and wheat [15]. However, only one mitochondrial genome from Caprifoliaceae (*Triosteum pinnatifidum*, NC_064333.1) has been sequenced to date. Therefore, the development of plastid markers for *N. jatamansi* would not only enhance genetic studies for its conservation but also provide marker sources for molecular investigations into other species within the Caprifoliaceae family. Ecological niche modeling (ENM), on the other hand, is pivotal for predicting species’ adaptive potential under climate change. For *N. jatamansi*, applying ENM would help to identify climate-resilient habitats and prioritize conservation areas, ensuring sustainable utilization of this endangered medicinal resource.

As of now, there are a total of 27 chloroplast genomes and one mitochondrial genome from the Caprifoliaceae available. Following the assembly of the organelle genomes of *N. jatamansi*, the evolutionary traits of its organelle genomes were examined. Comparative genomic analyses among Caprifoliaceae species were conducted to pinpoint highly variable regions that could potentially serve as markers for identification of Caprifoliaceae species. Finally, an ecological niche model for *N. jatamansi* was developed to offer insights into genetic conservation strategies for this species.

## 2. Materials and Methods

### 2.1. Chloroplast Genome Assembly and Annotation

*Nardostachys jatamansi* accession used for sequencing was collected from Hongyuan County (32°48′ N, 102°33′ E; altitude 3500 m), Sichuan Province, Chengdu, China, in July 2024. Voucher specimens (voucher no. NJ2024-SC-017) was deposited in Sichuan Academy of Grassland Sciences. The total genomic DNA of *N. jatamansi* was extracted from healthy leaves using the modified CTAB method [16]. The quantity and quality of DNA were assessed using a Nanodrop and 1% agarose gel electrophoresis, respectively. For short-read sequencing, the DNA was fragmented by sonication, followed by purification. Adenine was added at the end of the purified DNA fragments, and sequencing adapters were ligated. Subsequently, the fragment size selection with a range of 300–500 bp was performed via agarose gel electrophoresis, and the paired-end sequencing library with insert sizes of 150 bp was prepared through PCR amplification. The prepared library was sequenced on the Illumina Novaseq platform to generate raw sequencing data. After discarding adapter sequences and low-quality reads, the purified data were employed for the de novo assembly of the chloroplast genome of *N. jatamansi* using Spades v4.2.0 software [17] with kmer values defined at 55, 87, and 121. The chloroplast genome of *N. jatamansi* was assembled as a single circular structure and annotated in Geneious Prime vR11 [18] using *T. pinnatifidum* (NC_064333.1) as the reference. The resulting genome map was visualized with OGDRAW v1.3.1 [19] under default parameters. Annotation of chloroplast coding sequences (CDS) was conducted with prodigal [20], rRNA predictions were performed using hmmer v3.4 software [21], and tRNA prediction was conducted using aragorn [22] with default settings, respectively.

### 2.2. Mitochondrial Genome Assembly and Annotation

High-quality genomic DNA from *N. jatamansi* was purified for sequencing purposes followed by a plant genomic DNA kit provided by Tiangen Biotech Co., Ltd., Beijing, China. Long-read sequencing libraries were constructed utilizing the SQK-LSK109 ligation kit following manufacturer instructions and sequenced on an Oxford Nanopore device. Raw third-generation reads were filtered using filtlong software (https://github.com/rrwick/Filtlong, accessed on 20 May 2025) to exclude low-quality sequences with “--min_length 1000, --min_mean_q 20” parameters. Simultaneously, short-read Illumina data were used for hybrid construction.

The mitochondrial genome was assembled using reference plant mitochondrial core genes (https://github.com/xul962464/plant_mt_ref_gene, accessed on 20 May 2025) as seed sequences. Long reads were aligned with minimap2 v2.1 software [23] through an iterative procedure to elongate seed sequences and assemble full mitochondrial fragments. Corrected long reads were processed utilizing Canu v2.3 [24], and hybrid assembly with Illumina short reads was executed employing Unicycler v0.5.1 [25] with default settings. Given the presence of multi-circular conformations or the complex physical structure of non-circular shapes in the mitochondrial genome, the corrected third-generation sequencing data was aligned to the contigs with minimap2 v2.1 [23] and Bowtie2 v2.5.4 [26] software with default settings. Final contig visualization and structural validation were performed using Bandage v0.9.0 software [27], considering the existence of sub-genomic circular and intricate non-circular arrangements.

Gene annotation entailed comparing protein-encoding sequences and rRNA to published plant mitochondrial genomes using Geneious Prime vR11 [18], with manual refining to verify accuracy. Transfer RNAs (tRNAs) were discovered using tRNAscanSE v2.0.12 [28] with default parameters.

### 2.3. Codon Usage of the Organelle Genes

The relative synonymous codon usage (RSCU), GC content in the third position (GC3s), and effective number of codons (ENC) of each protein-coding gene in the organelles were calculated using codonW v1.4.2 software [29]. Subsequently, an ENC-plot analysis was conducted to establish the correlation between ENC and GC3s, with the fitted curve generated using the ggplot2 v3.5.2 R package [30].

### 2.4. Repeat Sequences Identification

In the current study, three categories of repeats were investigated: simple sequence repeats (SSRs), tandem repeats, and dispersed repeats. SSRs were identified using the misa v2.1 software [31] with the motif number of 1–6 base repeats set at: ≥10 repeats for mononucleotides, ≥5 for dinucleotides, ≥4 for trinucleotides, and ≥3 repeats for tetra-, penta-, and hexa-nucleotide motifs. Tandem repeats were detected by the trf software 4.09 (https://tandem.bu.edu/trf/trf.html, accessed on 25 May 2025) with “match 2, mismatch 7, indel 7, PM 80, PI 10, Minscore 50, MaxPeriod 2000” parameters, while dispersed repeats were detected with blastn [32] with “-evalue 1 × 10^−5^, -max_hsps 10, -word_size 7” settings. To visualized the repeat sequences, Circos v0.69.3 software [33] was employed.

### 2.5. Nucleotide Diversity and Nonsynonymous and Synonymous Analysis

Nucleotide diversity (Pi) serves to quantify the extent of nucleic acid sequence variation across different species, with highly variable regions potentially offering valuable markers for population genetics. In this context, the shared chloroplast gene sequences from 28 species within the Caprifoliaceae family were subjected to global alignment using the mafft v7.453 software [34] with default settings. Subsequently, the Pi values for each gene were computed utilizing DnaSP5 v5 [35]. Genes exhibiting substantial variation, as denoted by high Pi values, were visualized using MultiPipmarker [36].

The ratio of the non-synonymous mutation rate (ka) to the synonymous mutation rate (ks) is a crucial metric that sheds light on the types of selection processes at play. To investigate this in the present study, homologous gene pairs between two species were extracted and aligned utilizing the mafft v7.453 software. Subsequently, KaKs_Calculator v3.0 [37] was employed to determine the ka and ks values for each gene pair, employing the MLWL model.

### 2.6. Chloroplast and Mitochondrial Homology Sequence Analysis

The homologous sequences between the organelle genomes of *N. jatamansi* were identified through the employment of the BLAST v2.14.1 software [32], with the criteria of an E-value set at 1 × 10^−5^ and a minimum similarity threshold of 70%. Subsequently, these homologous fragments found between the chloroplast and mitochondrial genome underwent visualization using the Circos v0.69.3 software [33].

### 2.7. Species Distribution Model Construction

The data regarding the distribution of *N. jatamansi* was sourced from GBIF (https://doi.org/10.15468/39omei, accessed on 3 September 2025). The longitude and latitude coordinates of each specimen were leveraged to retrieve the corresponding climate data (bio1 to bio19, https://worldclim.org/) with the Arcgis v10.3 software [38]. Subsequently, pairwise correlation coefficients among these climate variables were calculated. If the pairwise correlation coefficients were higher than 0.8 or less than −0.8, the climate factors possessing the relatively less important biological functions were excluded to prevent potential overfitting. The significance of the retained environmental factors was determined through the application of Maxent v1.8.3 [39]. Using this method, distribution for *N. jatamansi* were developed for the current period, as well as for past period (Last Glacial Maximum, Mid-Holocene), and for future climate scenarios (2021–2040, 2041–2060, 2061–2080, and 2081–2100).

## 3. Results

### 3.1. Organelle Genome Sequencing, Assembly, and Annotation

A total of 1,420,093 and 4,262,797 clean reads were generated from Single Molecular Nanopore DNA and Illumina sequencing, respectively. The clean reads were utilized for the assembly of the organelle genomes of *N. jatamansi*, resulting in genome lengths of 1,229,747 bp and 155,225 bp for the mitochondrial and chloroplast genomes. The sequencing depth for the mitochondrial and chloroplast genomes of *N. jatamansi* was 173 X and 4119 X, respectively, with the formula Coverage = Total organelle-mapped bases/Organelle genome size. While the chloroplast genome could be accurately assembled into a single circular structure, the mitochondrial genome demonstrated a complex structure comprising 14 contigs (Figure 1). The chloroplast and mitochondrial genomes of *N. jatamansi* collectively encoded 129 and 79 genes, encompassing 84/42, 37/31, and 8/3 mRNA, tRNA, and rRNA genes, respectively.

### 3.2. Condon Usage Analysis of the Protein-Coding Genes (PCGs)

The organelle genomes of *N. jatamansi* utilize 63 codons to encode the 20 standard amino acids (Figure 2). The most common initiation codons are AUG and GUG. The relative synonymous codon usage (RSCU) values for each codon were computed, revealing that 32 codons have RSCU values greater than one within the organelle genomes (Appendix A). Notably, the codon AUG exhibits the highest usage frequency, with RSCU values of 6.98 and 3 in chloroplast and mitochondrial genomes, respectively.

To investigate factors contributing to codon bias beyond base composition, an ENC (effective number of codon)-plot analysis was conducted, with GC3s (GC content of the 3rd position of the codon) and ENC serving as the x- and y-axis, respectively. A higher ENC value suggests an even distribution of codon usage, indicating weak preferences. The ENC value of 61 for the chloroplast gene *psbF* suggests that its codons are used without any preference. The analysis revealed that many organelle genes deviate from or are distributed above the best-fit line, suggesting that these genes exhibit codon usage patterns that differ from the expected relationship between ENC and GC3s (Figure 2C,D). Particularly within the chloroplast genes, only *psbF*, *rpl23*, and *psbE* were positioned either above or in close proximity to the line. This observation indicates that the codon bias for these genes is predominantly influenced by selection pressure rather than mutation pressure alone, underscoring the combined impact of both factors on these specific gene sequences.

In the chloroplast genome of *N. jatamansi*, four types of repeat sequences—forward (F), reverse (R), complementary (C) and palindromic (P)—were identified (Appendix A), collectively accounting for 10,899 bp (7.02% of the chloroplast genome). Additionally, 177 simple sequence repeats (SSRs) were detected, with A/T, AT/TA, TTC, and ATAA being the most frequent repeats in monomer, dimer, trimer, and tetramer repeats, respectively (Figure 3A). These repeats were predominantly situated within the Large Single Copy (LSC) region, occupying 64% of the genome (Figure 3B,C), particularly in the intergenic regions. Notably, the genes *ycf1*, *ycf2*, *rpoC1*, and *ndhF* contained the highest number of SSRs (Appendix A). Moreover, the repeat sequences within the chloroplast genomes of 28 Caprifoliaceae species were also determined (Appendix A), which revealed that the *Leycesteria* sp. harbored the greatest total repeat length across the species, with 14,002 bp, followed closely by *N. jatamansi* with 10,899 bp. Among the Caprifoliaceae species, *N. jatamansi* exhibited the highest count of forward repeats (140) and palindromic repeats (84). Notably, a significant portion of these repetitive sequences were located in tRNA genes, with a smaller proportion present in protein-coding genes.

Abundant repeat sequences were detected in the mitochondrial genome of *N. jatamansi*, containing 309 SSRs, 256 tandem repeats, and 47,970 dispersed repeats (Appendix A). Notably, two pairs of large fragment dispersed repeats were identified, one located at chromosome 2 (4335 bp), and the other between chromosome 5 and chromosome 7 (3670 bp).

### 3.3. Sequence Polymorphism of Shared Chloroplast Genes Among the Caprifoliaceae

A total of 63 chloroplast genes were shared among the 28 Caprifoliaceae species, with four genes *rps19*, *rpl22*, *rpl20*, and *matK* exhibiting relatively higher Pi (nucleotide diversity) values (Figure 4, Appendix A). Except for eight species that showed limited variation compared with the reference species *Abelia chinensis*, the remaining species exhibited varying degrees of variation within these four genes. This suggests that these four genes could serve as potential markers for distinguishing between Caprifoliaceae species.

### 3.4. Nonsynonymous (ka) and Synonymous (ks) Analysis

To assess the impact of selective pressure on the protein-coding genes, pairwise nonsynonymous (ka) and synonymous (ks) values were calculated for the shared coding genes of Caprifoliaceae species (Figure 5). Among these genes, three chloroplast genes—*clpP*, *ycf1*, and *ycf2*—exhibited ka/ks values surpassing one in all pairwise comparisons. These findings suggest that these genes may be undergoing positive selection.

### 3.5. The Gene Transfers Between Chloroplast and Mitochondrial Genomes of N. jatamansi

The collinearity analysis identified a total of 71 homologous fragments between the chloroplast and mitochondrial genomes of *N. jatamansi* (Figure 6, Appendix A). These homologous fragments were primarily situated in genic regions within the chloroplast genome. In contrast, within the mitochondrial genome, the fragments displaying homology to the chloroplast genome were mainly distributed across intergenic or tRNA/rRNA regions. Notably, the chloroplast genes *ycf2* and *ycf1* underwent the most frequent gene transfer events. Additionally, the entire genic regions of five protein-coding genes—*rps4*, *petL*, *petG*, *rps19*, and *rps3*—exhibited homology to the corresponding mitochondrial regions.

### 3.6. Ecological Niche of N. jatamansi

Following the screening of correlation coefficients, six key climatic factors were selected to construct the niche model for *N. jatamansi* (Appendix A). The Maxent model highlighted elevation (44.7%) and bio 12 (24.8%, annual rainfall) as the most influential environmental variables. Analysis of the past, present and future niches of *N. jatamansi* revealed that historically and presently, its distribution has been relatively confined, primarily centered in the Qinghai–Tibet Plateau region of China (Figure 7A,B). The suitable range is anticipated to broaden, reaching Russia and South America from 2041 to 2060 (Figure 7C), followed by North America from 2061 to 2080 (Appendix A). However, a sharp reduction in the suitable range is predicted for 2081–2100, limited primarily to a few low-latitude regions worldwide.

## 4. Discussion

As a significant family boasting valuable crops and ornamental plants, the absence of genomic data for the Caprifoliaceae hinders research into genetic diversity and phylogenetic evolution within this group. Utilizing the conserved nature of plant chloroplast genomes, we conducted a comparative and analytical study of 28 Caprifoliaceae species, identifying high-variability regions that could serve as molecular markers for distinguishing among Caprifoliaceae plants. Additionally, we reported the second mitochondrial genome of the Caprifoliaceae family—*N. Jatamansi*, enriching the plant organelle genome database and offering insights into horizontal gene transfer. Notably, while *N. Jatamansi* is currently classified as an endangered species, as reflected in the past and present niche results, our model projections indicate that its distribution will significantly expand in the future.

Plant and animal mitochondria differ significantly in structural dynamics, with plant genomes exhibiting greater complexity [40]. In contrast to the relatively simplistic structure of animal mitochondrial genome, plant mitochondrial genomes exhibited complexity, featuring a variety of configurations including circular, linear, and sub-circular conformations. While the mitochondrial genome of *T. pinnatifidum* in Caprifoliaceae comprises a solitary circular structure [41], the mitochondria of *N. jatamansi* exhibit a principal circular form along with several sub-ring structures. Similar complexity exists in other species like *Indigofera amblyantha*, *Indigofera pseudotinctoria* [42], and *Aristolochia manshuriensis* [43], indicating this structural intricacy is common across plant taxa. Repetitive sequences in angiosperm mitochondrial genomes act as recombination hotspots, driving structural rearrangements and multi-conformation emergence [44]. In *N. jatamansi*, we detected 47,980 repeat pairs, vastly exceeding that found in *T. pinnatifidum* [41]. This includes two pairs > 1000 bp, 96 pairs ranged from 200 to 999 bp, and a remarkable abundance of short repeats (47,881 pairs), which is nearly 100 times higher than *T. pinnatifidum* (400 pairs). Studies have suggested that large repeats could arise from recombination events involving short repeats, and these short repeats are more prone to induce irreversible changes in the genome [40]. Consequently, we hypothesize that the combination of extensive large repeats and super-abundant short repeats in the mitochondrial genome of *N. Jamansi* may contribute to its structural complexity.

Another key driver of mitochondrial genome reorganization may be IGT, a phenomenon of broad genomic significance [45]. While IGT among chloroplast, mitochondrial, and nuclear genome is well-documented [42,46], its evolutionary implications—particularly in extremophile adaptation—remain actively explored. Given the limited genomic information available for *N. Jatamansi*, our research has focused on detecting IGT events specifically from chloroplast to mitochondrial DNA, aligning with current research on mitochondrial genome complexity as an evolutionary response to environmental challenges. Critically, although some studies suggest organelle-to-organelle IGT events may represent sequencing artifacts [42], our long-read sequencing approach provides unique advantages in distinguishing genuine transfers. Our analysis detected substantial chloroplast-derived sequences (49,575 bp across nine contigs) within the *N. Jatamansi* mitochondrial genome, including 13 homologous sequences > 1000 bp. These highly conserved transfers may fragment the mitochondrial genome, mirroring patterns observed in other angiosperms with complex, multi-conformation mitochondrial architectures [47]. Crucially, the coexistence of exceptionally abundant short repeats (47,881 pairs) and frequent IGT events provides a dual mechanism for recombination-driven structural diversification. We therefore propose that synergistic repeat proliferation and chloroplast-mitochondrial IGT underpin the intricate sub-genomic conformations in *N. jatamansi*, exemplifying how genomic fluidity contributes to plant mitochondrial evolution in demanding ecological niches.

According to the neutral theory of molecular evolution, the impact of base mutations within genes on codons is considered neutral or nearly neutral [48,49]. However, when codons within the genomes are influenced by selective pressures from the external environment, it can result in codon usage bias and alternations in base composition [50]. Codon bias is a significant aspect of genome evolution, shaped by a combination of natural selection, species-specific mutations, and genetic drift [51]. In a pattern exhibited by numerous species [41,51], the codons found in the chloroplast and mitochondrial genes in *N. Jatamansi* display a preference for A/U endings. This preference is speculated to be associated with RNA structure and secondary structure composition [52]. Moreover, the ENC values function as a metric for codon bias regarding the 20 amino acids, where values approximating 61 denote an even distribution of codon usage across each codon [51]. The ENC plot analysis revealed that all chloroplast genes of *N. Jatamansi* fell below the fitting line, with the exceptions of *psbF* and *rpl23*, suggesting that natural selection predominantly impacts chloroplast genes. Conversely, mitochondrial genes were positioned above the fitting line, signifying that these genes are influenced by base mutations in addition to natural selection. Given that genes related to photosynthesis and respiration are sensitive to environmental selection pressures, plastid genes typically experience significant levels of natural selection [53].

Our projection that *N. jatamansi*’s suitable habitat may expand toward Russia, South America, and later North America within the next 60 years aligns with broader biogeographic trends under climate change. Several studies suggest that high-elevation and cold-adapted species may shift poleward or to higher latitudes as temperatures rise [54]. The Qinghai–Tibet Plateau, currently its core habitat, is experiencing rapid warming [55], which could force range expansion into climatically analogous regions. Additionally, increased precipitation in northern latitudes under future climate scenarios may facilitate colonization in currently unsuitable areas [56]. However, the predicted contraction by 2081–2100 underscores potential range instability, consistent with studies showing habitat fragmentation in alpine species under prolonged warming [57]. Thus, while short-term expansion seems plausible, long-term survival may depend on dispersal capacity and conservation efforts in newly suitable regions.

## 5. Conclusions

This study presents the organelle genomes of a significant Tibetan medicine plant, *N. Jatamansi*, marking the second mitochondrial genome of the Caprifoliaceae family. Analysis of the comparative chloroplast genomes of Caprifoliaceae plants revealed regions of notable variation, such as *rpl22*, *rpl20*, *rps19*, among others, which hold potential as molecular markers for the differentiation of Caprifoliaceae species. ENC plot analysis indicated that in *N. Jatamansi*, chloroplast genes were primarily influenced by natural selection, whereas mitochondrial genes were impacted by micro-effective base mutations alongside natural selection. The high number of repeats in the mitochondrial genome of *N. Jamansi* and the extensive sequences derived from the chloroplast genome are likely contributing factors to the intricate structure of the mitochondrial genome of *N. Jatamansi*. The outcomes of the niche analysis revealed that altitude and annual rainfall were the primary climatic factors influencing the distribution of *N. Jatamansi*. Furthermore, the optimal distribution area of *N. Jatamansi* is anticipated to significantly expand in the next 60 years.

## Figures and Tables

**Figure 1 biology-14-01219-f001:**
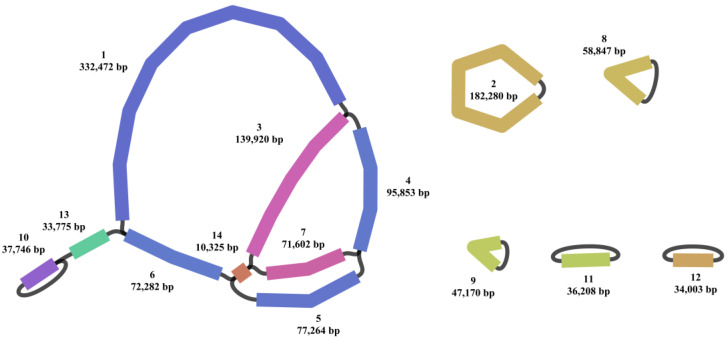
Mitochondrial genome map of *N. jatamansi.* In the Bandage visualization, the mitochondrial genome is color-coded and numerically labeled to distinguish its various regions.

**Figure 2 biology-14-01219-f002:**
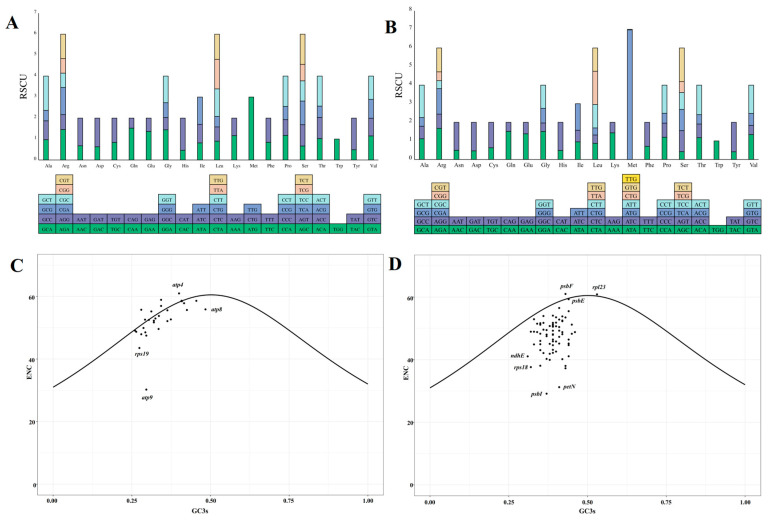
The relative synonymous codon usage and ENC-GC3s plots for mitochondrial (**A**,**C**) and chloroplast genes (**B**,**D**) of *N. jatamansi*. The lower section of panels A and B displays the types of codons for each amino acid in the organelle genomes, while the upper section shows the corresponding RSCU values for these codons.

**Figure 3 biology-14-01219-f003:**
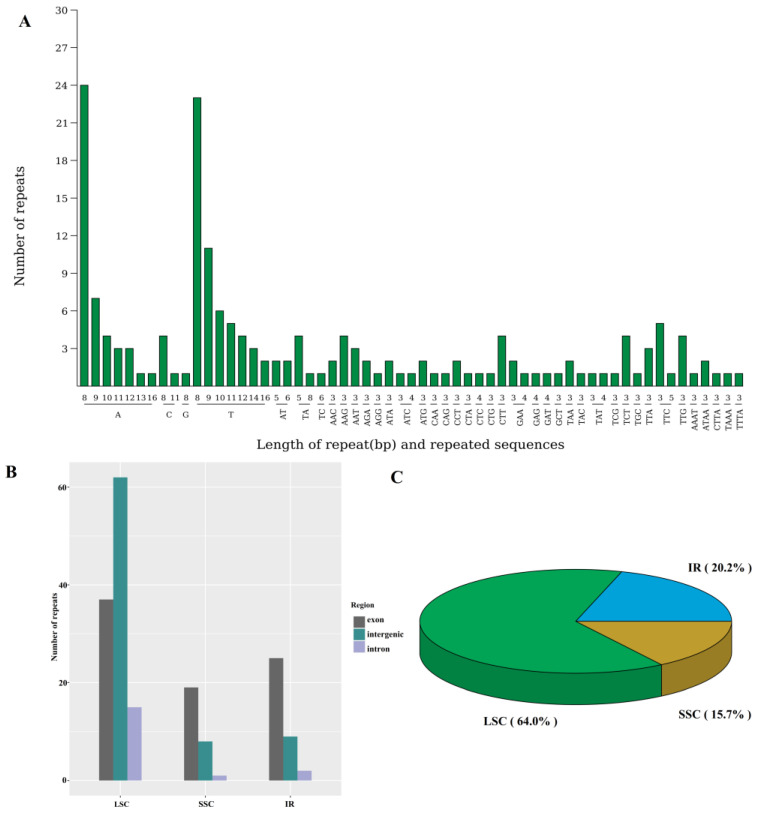
The simple sequence repeats within the chloroplast genome of *N. jatamansi*. (**A**), the number of each repeat motif; (**B**), the number of repeats in each region; (**C**), the ratio of the repeats. LSC, Large Single Copy region. SSC, Small Single Copy region. IR, Inverted Repeat region.

**Figure 4 biology-14-01219-f004:**
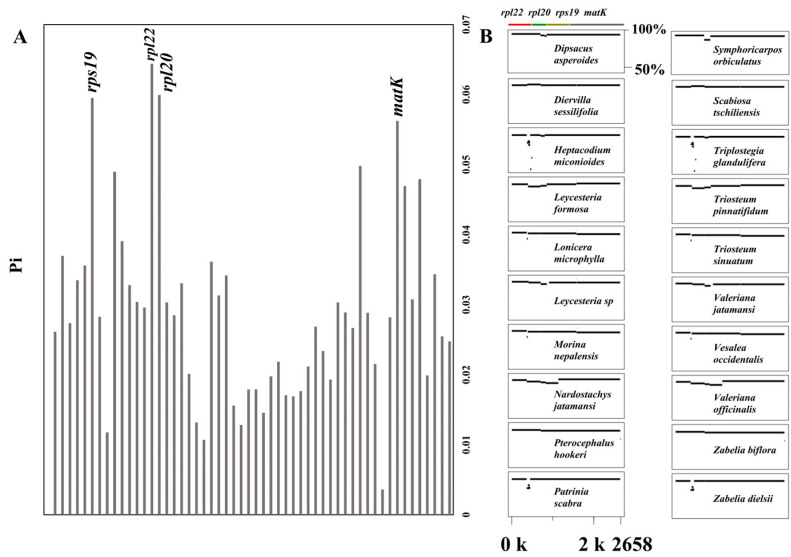
The nucleotide diversity (Pi) values of the shared chloroplast genes among the 28 Caprifoliaceae species (**A**). Detailed view from the MultiPipMaker alignment highlighting the sequence variations in the 20 Caprifoliaceae species exhibiting variants compared with the reference species of *Abelia chinensis* (**B**).

**Figure 5 biology-14-01219-f005:**
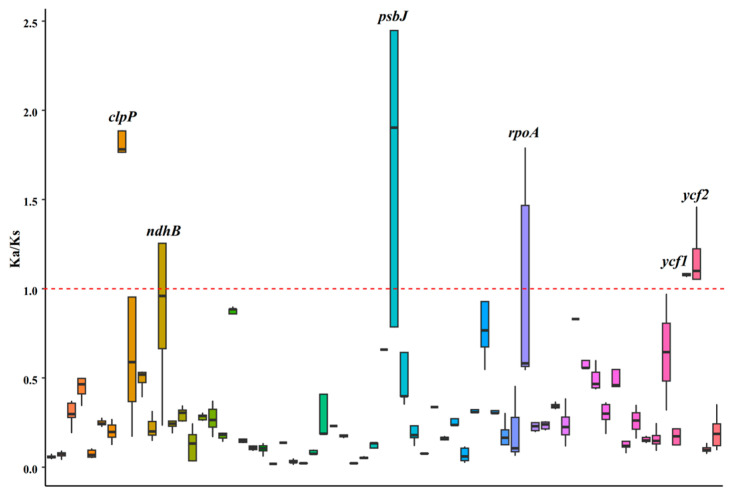
The nonsynonymous/synonymous (ka/ks) values of the shared chloroplast genes among the 28 Caprifoliaceae species. The red dashed line represents a Ka/Ks value of 1, which serves as a critical threshold for inferring selection pressure. The black line within each boxplot represents the median value of the distribution.

**Figure 6 biology-14-01219-f006:**
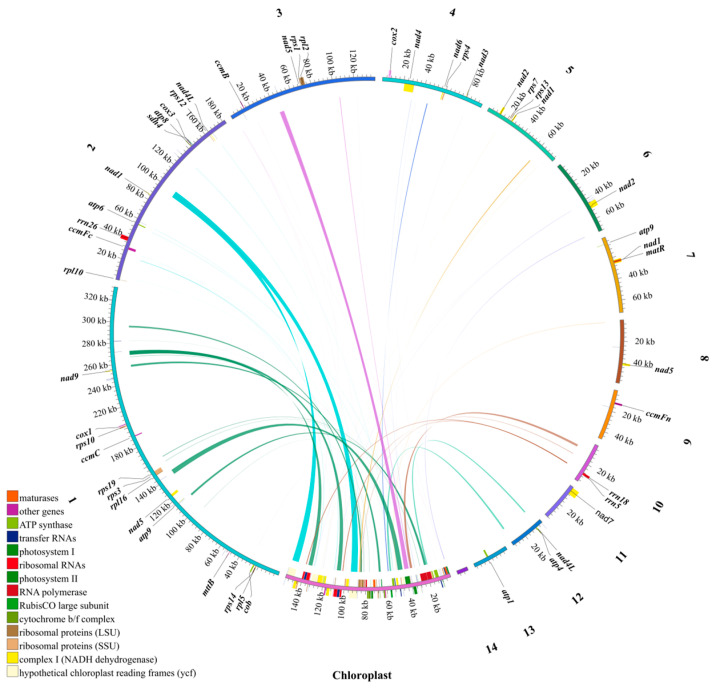
The homologous sequences between chloroplast and mitochondrial genome of *N. jatamansi.* The numbers outside represent mitochondrial DNA contigs.

**Figure 7 biology-14-01219-f007:**
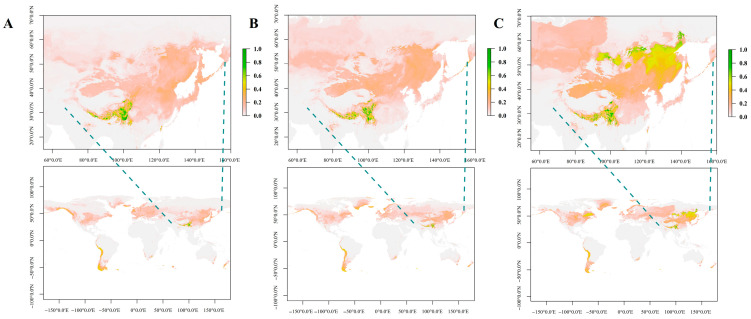
The ecological niches of *N. jatamansi* at past ((**A**), LGM), present (**B**), and future ((**C**), 2041–2060) periods. The areas shaded in green indicate greater suitability for the growth of *N. Jatamansi*.

## Data Availability

The sequencing clean data of chloroplast and mitochondrial genome of *N. Jatamansi* have been uploaded to China National GeneBank DataBase (https://db.cngb.org/) with the project number of CNP0005369, with reed IDs of CNR1100436 and CNR1100437.

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
