# Peer review of "Organelle Genomes of *Nardostachys jatamansi* Offer New Perspectives into the Evolutionary Dynamics of Caprifoliaceae"

_biology, 2025, doi:10.3390/biology14091219_

Round 1

Reviewer 1 Report

Comments and Suggestions for Authors

My comments are as follows:

1. Title: Organelle genomes of Nardostachys jatamansi offer new perspectives into the evolutionary dynamics of Caprofoliaceae

2. please make sure to uniform the format for numerical values, i.e., 47,980, instead of 47980. Please check throughout the manuscript.

3. There is no supporting discussion on the prediction of which the suitable habitat is expected to expand in the next 60 years. Considering that this is an important information for this endangered species, it is wise to provide evidence from literature that can support the findings obtained from the software.

Author Response

All requested revisions have been implemented, tracked in red text within the revised manuscript (Word version). As requested, continuous line numbering has been implemented in the revised manuscript.

  1. Title: Organelle genomes of Nardostachys jatamansi offer new perspectives into the evolutionary dynamics of Caprofoliaceae.

Answer: We have changed the title to “Organelle genomes of Nardostachys jatamansi offer new perspectives into the evolutionary dynamics of Caprofoliaceae”.

  1. please make sure to uniform the format for numerical values, i.e., 47,980, instead of 47980. Please check throughout the manuscript.

Answer: We have amended all values to the uniformed format.

  1. There is no supporting discussion on the prediction of which the suitable habitat is expected to expand in the next 60 years. Considering that this is an important information for this endangered species, it is wise to provide evidence from literature that can support the findings obtained from the software.

Answer: We sincerely appreciate your valuable comment. We have now added a discussion (Lines 349-359) citing relevant studies on poleward range shifts of alpine species under climate change (e.g., Chen et al., 2011; Parmesan & Yohe, 2003), increased precipitation at high latitudes (Walsh et al., 2020), and warming trends in the Qinghai-Tibet Plateau (Dullinger et al., 2019). These references substantiate our model's prediction of near-future expansion toward Russia and the Americas. Thank you for highlighting this important point.

Reviewer 2 Report

Comments and Suggestions for Authors

Dear Authors,

I have had the opportunity to review your manuscript entitled "Organelle genomes of Nardostachys jatamansi offering new perspectives into the evolutionary dynamics of the Caprifoliaceae family." Your work presents a significant contribution to the field of plant genomics and evolutionary biology, particularly within the Caprifoliaceae family. The study's dual focus on chloroplast and mitochondrial genomes is both timely and relevant, offering new insights into structural complexity, molecular evolution, and phylogenetic diversity.

The identification of high-variability regions within the chloroplast genome and the characterization of extensive repeats and intergenomic transfer in the mitochondrial genome provide valuable resources for future molecular marker development and evolutionary studies. Furthermore, the comparative approach adopted in your study enhances our understanding of organelle genome dynamics in an endangered yet scientifically important species.

While the manuscript demonstrates scientific merit, certain areas—particularly in terms of clarity, methodological detail, and synthesis of discussion—would benefit from further revision. The accompanying attachment contains section-by-section reviewer comments to help strengthen the manuscript and support its potential for publication.

I commend your efforts in advancing the genomic understanding of N. jatamansi and encourage you to consider the comments provided to enhance the clarity, rigor, and impact of your study.

Sincerely,

Author Response

All requested revisions have been implemented, tracked in red text within the revised manuscript (Word version). As requested, continuous line numbering has been implemented in the revised manuscript.

  1. The abstract would benefit from enhanced coherence. The information is presently conveyed in a compact fashion lacking clear transitions. Consider structuring the abstract into coherent subsections or more distinct theme categories.

Answer: We have polished the entire abstract and restructured its logical flow to enhance its fluency.

  1. The term "for species distinguish" should be replaced by "for species distinction" or "for distinguishing species”. In addition, when you write something like "The intracellular gene transfer analysis documented several fragments and six genes from the chloroplast genome," you should not use sentence fragments. Find out if these genes were functionally linked. The last statement talks about a base for investigating medicinal qualities, however it does not make a clear connection between the genomic findings and the pharmacological traits. A single sentence outlining how the genomic data could help with this kind of research would make the conclusion stronger. The following are suggestions for rewriting the abstract part of your case scenario.

Answer: Thanks for your suggestion. We have amended "for species distinguish" to " for identification of species", and amended the whole Abstract section according to your rewriting suggestions. This revised abstract now effectively conveys the methodological depth and scientific significance of our study.

  1. You give numbers (such as repeat count and genome sizes), but you may also briefly talk on what these results mean for biology. For example, how does the number of repeats in N. jatamansi compare to that of other similar species?

Answer: We have discussed this number variation among N. jatamansi and its related species such as T. pinnatifidum, whose abundance of repeats is much fewer, which may give raise to the structural complexity of the mitochondrial genome of N. jatamansi (Lines 302-312, Lines 326-331).

Introduetion section:

  1. Some sentences are too long and hard to understand. Think about dividing them up into smaller, clearer pieces. For example, the statement that starts with "The essential metabolic processes involved in water uptake" could be split up to make it clearer.

Answer: We have polished and simplified all similar expressions to enhance conciseness and readability.

  1. There are several times when the medicinal uses of N. jatamansi are talked about in depth. To prevent repeating, these can be summed up in a short paragraph.

Answer: We have summarized the medicinal properties of N. jatamansi and eliminated redundant content.

  1. Explain how organelles and conservation are related. The study's goal is to help conservation efforts; however, it would be helpful to explain how organelle genome data helps with this goal.

Answer: The organelle genomes provide valuable insights for molecular marker development and genetic research, particularly in species lacking nuclear genome assemblies like N. jatamansi. This rationale has been articulated in the Introduction (Lines 67-79).

  1. What you need to know about ecological niche modelling. You should talk about the idea and importance of ecological niche modelling earlier, so that adding it feels more related to the genomic work.

Answer: Ecological niche modelling (ENM) is pivotal for predicting species' adaptive potential under climate change. For N. jatamansi, applying ENM would help to identify climate-resilient habitats and prioritizes conservation areas, ensuring sustainable utilization of this endangered medicinal resource. We have added related description in the Introduction section (Lines 79-82).

Methodology section

Section 2.1-Chloroplast Genome Assembly and Annotation

  1. Lack of Specificity in DNA Extraction: The CTAB approach is acknowledged but neither cited or described. Was it modified or standard? Include any relevant references or protocol variations.

Answer: We have incorporated a citation for the optimized CTAB-based DNA extraction protocol in the revised manuscript.

  1. Quality Control Metrics That Are Not Present: No mention is made of: DNA quality (for example, the findings of gel electrophoresis or nanodrop analysis). Sequencing read length (e.g., 150 bp paired-end?), Total number of reads generated or genome coverage (depth).

Answer: We appreciate this insightful comment. DNA quality assessment via Nanodrop and and 1% agarose gel electrophoresis analysis has now been incorporated into the Methods section (Lines 96-97). The Illumina PE150 library preparation protocol has been incorporated in Lines 99-102 to ensure reproducibility of sequencing. The genome coverage (depth) of both chloroplast (173 ×) and mitochondrial (4,119 ×) genome of N. jatamansi are given in the result section (Lines 186-188).

  1. Uncertainty in the Process of Size Selection: The statement about fiagment size selection using agarose gel electrophoresis might be improved by providing the target size range (e.g., 300-500 bp)

Answer: We have incorporated these details at Line 100 of the revised manuscript.

  1. There is no information on how the assembled genome was checked (for example. by mapping back reads or confirming circularity), and there are no assembly metrics like N50, number of contigs, etc. A lot of studies involve visualization or manual checking after annotation, like with Geneious or Artemis. Think about saying if this step was done.

Answer: The chloroplast genome of N. jatamansi was assembled as a single circular structure and annotated in Geneious Prime vR11 using T. pinnatifidum (NC_064333.1) as the reference. The resulting genome map was visualized with OGDRAW v1.3.1 under default parameters. These methodologies are detailed in Lines 105-108.

Section 2.2-Mitochondrial genome assembly and annotation

  1. The part is too deep and technical, listing tool names and actions in a way that is too repetitious or unnecessary, without breaking them up into logical stages like preprocessing, assembly, and validation. There are a lot of situations when phrases like "third-generation sequencing data” are used. The part is hard to follow because the phrases are so long. You can cut and reorganize the section into three distinct paragraphs: DNA Extraction and Sequencing, Assembly Strategy, and genome Visualization and Annotation as below.

Answer: We sincerely appreciate your insightful suggestions regarding methodological clarity. The Genome Assembly section has been comprehensively restructured as recommended.

  1. There is no explicit indication of the annotation pipeline, and the stages for annotation are only loosely described as "manual adjustment." For example, was any automated tool utilized before manual curation? No Assembly Metrics Given: There are no important results like the number of contigs, N50, coverage, or total size before and after filtering. There is no information about the sample's origin, DNA concentration, platform name (such ONT PromethION or MinION), or read length stats.

Answer: We have incorporated the relevant additions throughout the manuscript at Lines 106-109, 113-119, 128-130, and 183-192.

  1. But some parts need to be improved for clarity and effect. First, the language used in the discussion should be clearer, certain ideas are repeated or too comprehensive. which could make the main points less clear. For example, it would be easier to summaries the part about repeat abundance and how it compares to T pinnatifidum. The codon usage analysis portion is also useful, although it might have more of an effect if it were more directly tied to functional implications. Also, while IGT is important, it would be more useful if the findings were connected to broader evolutionary or ecological issues, such as how organisms adapt to living at high altitudes. Finally, bringing up current research on the complexity of plant mitochondria and the evolution of genomes could help make the background more solid.

Answer: We sincerely appreciate your insightful suggestions for improving the clarity and impact of our discussion. We have carefully revised the manuscript to address each point raised (Lines 303-331).

Reviewer 3 Report

Comments and Suggestions for Authors

Dear Authors,

Thank you for the interesting manuscript. It is of special interest to me as I am working on assembly of mitochondrial genome at the moment.

Maybe this is unfortunate for you as I was very interested in the methods section of your work to see and compare what software you used and what the settings were. I saw that you did not include almost any information at all regarding the settings used for nearly each software used. As one of the goals of any scientific publication is to transparently show the quality of your work, the description should be sufficient to allow other scientists to reproduce your results. This means that the settings of the software used should be indicated in detail. I suggest either to include this information in the methods section or to create a supplementary file listing all the software and respective settings. You could also benefit from providing a visual depiction of the analysis pipeline either in the text of the manuscript or as an appendix.

Please also indicate kits used and indicate catalogue numbers and whether you followed manufacturer's instructuions or modified the protocols.

I was also sad to see that the manuscript does not contain page lines. This made it complicated for me to provide you with feedback.

Other remarks.

Materials and methods

section 2.1.

In this section mention what DNA fragment size did you select for.

full settings for the software.

section 2.2.

Why did you fragment DNA if using long read sequencing (1st sentence)?? Please explain, makes no sense to me. You yourself tell in next sentence that you select for large fragments.

What does gelation mean in this context, what method specifically do you have in mind?

What method was used for DNA library quantification?

Candidate sequences? Candidate for what? For which process?

Software settings missing.

Section 2.4. Clarify the second sentence - meaning of the numbers 10, 5, 4, 3, 3 and 3.

Section 2.5. & 2.7.  - software settings.

Results

In the first sentence you refer to a specific number of reads. Are these reads of total genomic DNA? Then how do you calculate coverage (173 X and 4419 X)? Please explain.

First sentence of section 3.3. does not explain the meaning of label "C" in the Fig. S1

Figure 3 is too small to read in the present form.

Fig. S2 has no labels or explanations.

Elements A and B of Figure 4 lack detail (rest of gene names in A, species names in B).

Figure 6. Indicate that the numbers represent mitochondrial DNA contigs.

If possible, make Figure 7 a bit larger.

for Discussion

Speaking about IGT from organelle to organelle, some sources say that this is a rarre occurance and that sometimes this is a result of errors in sequencing data analysis. I thing you should reflect on this, specifically as you, at least partially, have used long read sequencing which should decrease the probability of data showing IGT being generated due to error. You could use this to your advantage in the discussion.

Data availability statement 

when looking for the accession number provided in your data availability statement, a project "Plastid genome sequencing rawdata of Bistorta viviparum" is found. That might be confusing. Please mention that, within the accession CNP0005369 one has to look for reed IDs CNR1100436 and CNR1100437

Comments on the Quality of English Language

Abstract

Change "distinguish" with the appropriate word or change the sentence.

indicate clearly that you are speaking about gene transfer between organelles to avoid exclude any readers to thing that you are speaking about organelle - nucleus or vice versa transfer.

Introduction

Avoid statements involving pseudoscience (you refer to regulation of Qi).

"As a consequence... " should be used in a sentence following reference 7.

"China", not "Chinese" should be used in next line.

Materials and methods

section 2.1. third row. "were", not "ware"

section 2.2.

"Flow" doesn't need a capital letter.

Third-generation, not three-generation.

Sentence following reference 24 - what do you mean by presence of multiple cycles?

Conclusions

review the 1st sentence. a word is missing.

Author Response

All requested revisions have been implemented, tracked in red text within the revised manuscript (Word version). As requested, continuous line numbering has been implemented in the revised manuscript.

  1. Maybe this is unfortunate for you as I was very interested in the methods section of your work to see and compare what software you used and what the settings were. I saw that you did not include almost any information at all regarding the settings used for nearly each software used. As one of the goals of any scientific publication is to transparently show the quality of your work, the description should be sufficient to allow other scientists to reproduce your results. This means that the settings of the software used should be indicated in detail. I suggest either to include this information in the methods section or to create a supplementary file listing all the software and respective settings. You could also benefit from providing a visual depiction of the analysis pipeline either in the text of the manuscript or as an appendix. Please also indicate kits used and indicate catalogue numbers and whether you followed manufacturer's instructions or modified the protocols. I was also sad to see that the manuscript does not contain page lines. This made it complicated for me to provide you with feedback.

Answer: We sincerely appreciate your insightful feedback regarding methodological transparency. Detailed settings for all bioinformatics tools are now fully documented in the manuscript.

Materials and methods

  1. section 2.1. In this section mention what DNA fragment size did you select for. full settings for the software.

Answer: Comprehensive details regarding DNA fragment size selection protocols and all software parameter configurations are provided.

  1. section 2.2. Why did you fragment DNA if using long read sequencing (1st sentence)?? Please explain, makes no sense to me. You yourself tell in next sentence that you select for large fragments. What does gelation mean in this context, what method specifically do you have in mind? What method was used for DNA library quantification? Candidate sequences? Candidate for what? For which process? Software settings missing.

Answer: DNA fragmentation was performed only for Illumina short-read librarieswhich used for chloroplast genome assembly and hybrid assembly validation and error correction of mitochondrial genome. While long-read libraries (Nanopore) maintained DNA integrity following standard SQK-LSK109 protocols. Algorithmic details of mitogenome assembly such as core sequences elongation are described. We have now clarified related information in the revised manuscript.

  1. Section 2.4. Clarify the second sentence - meaning of the numbers 10, 5, 4, 3, 3 and 3. Section 2.5. & 2.7. - software settings.

Answer: The numbers 10, 5, 4, 3, 3 and 3 represent the motif number of 1-6 base repeats set at: ≥10 repeats for mononucleotides, ≥5 for dinucleotides, ≥4 for trinucleotides, and ≥3 repeats for tetra-, penta-, and hexa-nucleotide motifs, respectively. We have addressed similar software settings in the revised manuscript.

Results

  1. In the first sentence you refer to a specific number of reads. Are these reads of total genomic DNA? Then how do you calculate coverage (173 X and 4419 X)? Please explain.

Answer: These sequences represent clean reads derived from post-filtering processing of raw sequencing data. The genome coverage is calculated based on the formula: Coverage = Total organelle-mapped bases / Organelle genome size. We have supplemented these details in the main text.

  1. First sentence of section 3.3. does not explain the meaning of label "C" in the Fig. S1

Answer: The designation "C" denotes complementary repeats. We have now incorporated this classification in the manuscript.

  1. Figure 3 is too small to read in the present form. Fig. S2 has no labels or explanations. Elements A and B of Figure 4 lack detail (rest of gene names in A, species names in B). Figure 6. Indicate that the numbers represent mitochondrial DNA contigs. If possible, make Figure 7 a bit larger.

Answer: We have comprehensively addressed the reviewer's figure concerns: Figure 3 and 7 were redesigned at higher resolution (600 dpi) with enlarged elements and optimized layouts to ensure visual clarity; Figure S2 now includes a descriptive legend in Supplementary Materials Section; Figure 4B incorporates expanded species names, while Figure 4A gene-level details are now fully documented in the newly added Table S4 (containing all 63 gene names with corresponding Pi values) due to spatial constraints; Figure 6 now explicitly annotates mitochondrial DNA contigs (1-14) in its legend; and Figure 7 was enlarged with the addition of an inset panel magnifying key structural details to enhance readability.

Discussion

  1. Speaking about IGT from organelle to organelle, some sources say that this is a rare occurrence and that sometimes this is a result of errors in sequencing data analysis. I thing you should reflect on this, specifically as you, at least partially, have used long read sequencing which should decrease the probability of data showing IGT being generated due to error. You could use this to your advantage in the discussion.

Answer: As suggested, we have significantly strengthened our discussion (Lines 313-331) to explicitly leverage the advantages of long-read sequencing in eliminating IGT false positives.

Data availability statement

  1. when looking for the accession number provided in your data availability statement, a project "Plastid genome sequencing rawdata of Bistorta viviparum" is found. That might be confusing. Please mention that, within the accession CNP0005369 one has to look for reed IDs CNR1100436 and CNR1100437

Answer: This information has been incorporated into the Data Availability Statement.

Comments on the Quality of English Language

  1. Abstract: Change "distinguish" with the appropriate word or change the sentence. indicate clearly that you are speaking about gene transfer between organelles to avoid exclude any readers to thing that you are speaking about organelle - nucleus or vice versa transfer.

Answer: We have revised "species distinguish" to "species identification" and clarified "intracellular gene transfer" as specifically denoting inter-organelle transfer (between chloroplast and mitochondria), not organelle-nuclear transfers.

  1. Introduction: Avoid statements involving pseudoscience (you refer to regulation of Qi). "As a consequence... " should be used in a sentence following reference 7. "China", not "Chinese" should be used in next line.

Answer: We have revised "regulation of Qi" to "metabolic regulation" for scientific precision. Additionally, the conjunction " As a consequence..." was inserted following Reference 7 (Line 64) to strengthen logical flow, and "Chinese" was replaced with "China".

  1. Materials and methods: section 2.1. third row. "were", not "ware". section 2.2. "Flow" doesn't need a capital letter. Third-generation, not three-generation. Sentence following reference 24 - what do you mean by presence of multiple cycles?

Answer: All spelling errors have been corrected, and "multiple cycles" has been replaced with the precise terminology "multi-circular conformations" to accurately describe mitochondrial genome architecture.

  1. Conclusions: review the 1st sentence. a word is missing.

Answer: We have clarified the species' botanical identity by replacing "medicine" with "medicinal plant" throughout the manuscript.

Round 2

Reviewer 3 Report

Comments and Suggestions for Authors

Thank you for addressing my concerns! Good luck in further studies!

Author Response

Dear Reviewer,

Thank you very much for your kind words and for your insightful comments, which have significantly improved our manuscript.

We truly appreciate your time and effort.

Sincerely,

Xiong Lei